Design and research of automated warehouse simulation platform based on virtual visualization framework

Di Huajun 40551445@ncepu.edu.cn
Department of Finance and Economics, Shanxi Engineering Vocational College , Taiyuan, Shanxi , China
Kong Xiangjie
Electronic publication date: 2024 Jan 22
Publication date: 2024
Volume: 10
Electronic Location ID: e1809
Received 2023 May 12; Accepted 2023 Dec 18
Copyright: © 2024 Di
Copyright year: 2024
Copyright holder: Di
License: This is an open access article distributed under the terms of the Creative Commons Attribution License, which permits unrestricted use, distribution, reproduction and adaptation in any medium and for any purpose provided that it is properly attributed. For attribution, the original author(s), title, publication source (PeerJ Computer Science) and either DOI or URL of the article must be cited.
License URL: https://creativecommons.org/licenses/by/4.0/

Keywords: Automated warehouse, Warehouse management system, Virtual platform, Simulation test, System control

Funding: Research on the Construction Path of Modern Logistics Management Industry-Education Integration GH-220471 This study was funded by the “Research on the Construction Path of Modern Logistics Management Industry-Education Integration Training Base in Higher Vocational Colleges under the Background of Comprehensive and High-quality Development (GH-220471).” The funders had no role in study design, data collection and analysis, decision to publish, or preparation of the manuscript.

==============================
The automated storage and retrieval system (AS/RS) has been developed rapidly and has been widely used in various fields. To achieve the ultimate goal of improving the efficiency of warehouse operation, this article studies the problem of goods location- allocation on the premise of a comprehensive analysis of the efficiency of automated warehouses and the overall stability of shelves. The mathematical model is established. Traditional Tabu search algorithms depend on Tabu table construction to avoid multiple continuous cycles that may occur in the direct search process. In this article, the improved algorithm can directly search and store in the Tabu table to prevent the algorithm from infinite search. A complete solution in that Tabu list continues to be passed to act as a point solution to replace the iteratively update product until a set iteration interval of the product is completed. The machine learning system can get the optimal solution. The improved algorithm is applied to solve the model, and the running results and the performance of the algorithm itself are obtained. The improved Tabu Search algorithm has more advantages in the process of solving the model because it reduces the size of the whole problem. The algorithm also uses the penalty optimization function as a measure of each solution. The greater the penalty is, the worse the solution is, and the smaller the penalty is, the better the solution is. The operation process of the material in and out of the warehouse in one day is studied, and the validity of the optimization scheme of goods location allocation is verified by substituting specific data for the solution. Finally, the case of goods in and out of the warehouse is simulated by using simulation software, and the running state and efficiency of the stacker are displayed according to the data. The Tabu direct search table is combined with the structure and function of the principle of amnesty direct search to achieve the goal of direct search and decision-making. The equilibrium coefficient of its operation is calculated by using the data, which proves that the scheme is feasible and has practical significance and research value.

Introduction

With the rapid progress of information technology and the proposal of the concept of Industry 4.0, the automated warehousing system is about to enter the era of intelligence, which gives new requirements to the automated warehousing system, that is, we should not only upgrade the hardware equipment iteratively but also pay more attention to the research of supporting software and algorithms (Yang, Wu & Ma, 2021). In the past, the software system supporting the automated warehouse was often developed by foreign companies. As more and more domestic companies began to notice the importance of the software systems, they began to develop their supporting systems, such as warehouse management systems (WMS), warehouse control systems (WCS), etc., (Rosa, 2022). There are many hidden dangers and low efficiency in the daily operation of logistics warehousing, but the increasing production demand makes the warehousing capacity unable to keep up with the production demand. In view of the existing problems in the warehouse, this article combines the advantages of modern logistics technology and automated warehouse. From the point of view of computers and software, continuous research and development of computer systems and core decision-making algorithms of the automated warehouses, replacing manpower with a computer, will liberate more labor force, reduce human accidents and improve the efficiency of the warehouse. In today’s advocacy of energy conservation and industrial upgrading, it is of practical significance to study the software system and algorithm of intelligent warehouses (Liu & Yang, 2022).

In recent years, the automated warehousing system has shown a trend of diversification. In addition to the traditional stacker warehouse, many new high-density warehouses have emerged, such as the high-density warehouse using shuttle cars, the automated warehouse using artificial intelligence AGV cars, and so on. As the design of warehouse shelves and the equipment used are different, many different warehousing solutions are derived. The software systems and algorithms used in different warehousing solutions are not the same, which requires researchers to conduct targeted research (Lei et al., 2020).

Sadiq, Landers & Don Taylor (1996) introduced the storage strategy-dynamic location planning to solve the problem of more and more miscellaneous goods in the warehouse, and this strategy is also relatively applicable to products with shorter life cycles (Yuxin et al., 2023). Van den Berg & Zijm (1999) used time parameters and dynamic programming methods to study the problems related to goods warehousing. At the same time, because the warehousing cost of each link is different, different cost expenses are combined with the operation process for systematic analysis (Zhou et al., 2020); Rouwenhorst et al. (2000) found through research that the warehousing efficiency will reach the highest level for warehousing goods less than or equal to 10 categories (Tripicchio, D’Avella & Unetti, 2022). Hsieh & Tsai (2001) proposed the location-allocation scheme of parts and components related to the product BOM information table. Muppani & Adil (2008) constructed a dynamic programming model to analyze all the goods on the automated shelves and determine the relationship between them in this way. However, this method is not suitable for automated warehouses with a large number of raw materials. At the same time, it also takes into account the cost of picking operations and storage space (Truong, Dang & Nguyen, 2020). In 2016, Miguel Horta used mathematical optimization technology (based on the least square method) to plan the layout of the warehouse, so that the automated warehouse can complete the distribution work in real-time (Truong, Dang & Nguyen, 2017).

At present, there is little research on virtual and visual intelligent warehouses. The intelligent warehouse studied in reference (Truong, Dang & Nguyen, 2018) is similar to the content of this article. The proposed model for allocating multi-objective warehouse storage has the characteristics of simplicity and efficiency. However, there are still some shortcomings in the above research literature: most of them are based on the high physical straight-line distance between the warehouse storage and the entrance of the channel, and whether the warehouse storage must be evenly distributed between the layers. However, for deeper optimization objectives, such as the height of the warehouse storage layer, how to distribute evenly when multiple orders are completed, and how to calculate the workload of equipment requirements still need to be researched. A more balanced distribution of tasks and objectives needs to be established.

Due to the limitation of a single algorithm, in recent years, there are more and more applications of hybrid algorithms with complementary optimization ability, and its design and analysis is also an important topic of algorithm research. Based on the multi-channel block layout environment in the logistics warehouse, it is necessary to provide a scientific decision for the order batching optimization under this warehouse layout (Xuejiao & Wennan, 2019). In this article, a mixed integer programming model is established to solve the material storage management problem under multi-block warehouse layout, which combines the characteristics of multi-channel and multi-block, and takes the minimization of the total picking distance as the objective. In this article, a mixed integer programming (MIP) model based on MVC (model-view-controller) is proposed to solve the material storage management problem under multi-block warehouse layout, which combines the characteristics of multi-channel and multi-block and minimizes the total picking distance. The hybrid strategy of improving the population by Tabu Search is to perform a global search first. To improve the quality of the individuals in the population, the individuals in the population have a larger distribution range in the solution space area, and a Tabu Search is performed on the individuals in this population (Liu et al., 2023). According to the above problems, this article uses the virtual scene to restore the structure of the automated warehouse and simulates the entire workflow of the automated warehouse, designs a three-dimensional virtual simulation platform for the automated warehouse, uses an improved Tabu Search algorithm to solve, and obtains the location-allocation scheme. The effectiveness and superiority of the improved Tabu Search algorithm are verified by comparing the results, which proves the effectiveness of the research content in this article.

Related work

Warehouse conveying control system

The warehouse transportation control system uses a scheduling mode, which will detect the movement of the equipment in real-time and constantly feed back the movement in the process of transportation. The system adopts an automatic control mode. In this way, the virtual host computer sends instructions to the controller of the equipment. After receiving the instructions, the controller automatically controls the equipment to complete the operation and feeds back the motion state of the equipment to the virtual host computer. This part of the system is the connecting bridge between the automated finished product palletizing system and the automated warehouse system.

In this virtual simulation scenario, high-rise shelves will be used to allow goods to enter and exit automatically and improve work efficiency. In this article, the stacker is used to access the goods, and the warehouse area and the storage area are connected by the roller conveyor line. The automated warehouse is mainly composed of a roller conveyor line, stacker, shelf, mechanical arm, AGV car, and other parts. The working mode of the automated warehouse is as follows: when entering the warehouse, a worker first places the goods on a roller conveying line, and the roller conveying line sends the goods to a warehouse area. Then, a stacker takes down the goods from the roller conveying line, and finally, the stacker transports the goods and places the goods in a designated bin. When going out of the warehouse, the stacker takes out the goods in the designated bin, then places the goods on the roller conveyor line and sends them to the automatic sorting area. The identification device of the automatic sorting system identifies which lane the goods should belong to through the bar code, the conveyor will send the goods to the corresponding lane. Then, the mechanical arm will put the goods neatly on the pallet. Finally, the AGV car delivers the goods and pallets to the goods storage area (Yan, Himan & Yangkun, 2020). The overall planning of the automated warehouse virtual simulation is shown in Fig. 1.

Figure 1 Overall planning of warehouse virtual simulation.

Overall architecture of simulation system

The 3D models of shelves, stackers, conveyor belts, mechanical arms, pallets, automatic sorting equipment, and AGV cars in the automated warehouse are built by 3DMAX. Then, the 3D models of the automated warehouse are imported into the engine for rendering and mapping. Actions are added to the stacker, cargo, AGV trolley, and mechanical arm in the three-dimensional model of the automated warehouse to present the virtual simulation design more perfectly. The overall framework is shown in Fig. 2.

Figure 2 Overall framework diagram.

By creating, accessing, reading, and writing the shared memory, the variables in the PLC ladder diagram are analyzed and assigned to the three-dimensional model of the automated warehouse. The purpose is to control the access of the automated warehouse through the PLC ladder diagram, which is more in line with the realistic operation mode (Xin, Jinsong & Junlin, 2019).

Overall composition of storage system

In this article, through the three-dimensional simulation of the physical unit of the logistics warehouse, the complete logistics system is shown. By the reliable visualization means for the design and planning of the logistics system, the program is improved. A complex multi-objective optimization system will run with the help of various parameters. Then the output of statistical results for decision makers of reference is obtained.

Overall composition of the warehousing system

In this article, the composition of the automated warehouse is redesigned, which can make the warehousing more rapid and accurate. The system mainly includes four aspects: the automated warehouse system, the automated finished product palletizing system, the conveying system, and the system controller. The warehousing system framework is shown in Fig. 3.

Figure 3 Storage system framework.

In Fig. 3, two-way data communication can be realized between the automated warehouse and each component to realize the timely feedback of data. For example, after receiving the instructions from the conveying system, the conveyor can feed back the data from the start to the completion of the goods conveying.

Overall scheme of control system

Whether the whole system runs smoothly and efficiently depends on the design of the control system. The control system needs to be designed according to certain characteristics of the goods and the frequency of goods access. It should conform to the actual production operation, which not only realizes the required system as a whole but also improves the overall work efficiency so that the system can be more in line with reality (Longman Ryan et al., 2023). The control of the system mainly includes the control of the automatic palletizing system, automated warehouse system, and storage and transportation system. The following is an introduction to these three aspects: (1) Automatic palletizing control system

The control of the automatic finished product palletizing system mainly includes: the controller controls the palletizing robot to complete the actions that should be completed, and the goods are neatly stacked on the pallet or conveyor belt (Zhang & Lv, 2022). The controller controls the conveyor belt, and its main function is to transport the goods. The controller detects the operation of the goods on the conveyor belt. Location detects whether the goods arrive at the place where they should arrive (Changpu & Yichun, 2019). The virtual host computer sends job instructions to each part. The controller controls the relevant equipment, and all aspects of each equipment are fed back to the virtual host computer. The structure of the automatic palletizing control system is shown in Fig. 4.

Figure 4 Structure block diagram of automatic palletizing control system.

(2) Automated warehouse system

The automated warehouse system mainly controls the movement of the stacker. The correct movement of the stacker is the most critical part of the project. The control of the stacker mainly includes receiving operation instructions and specifying whether the goods are warehoused or warehoused (Ammouri, 2023). If the control design of the stacker is reasonable, the overall work efficiency will be improved, mainly including the control of vertical lifting, horizontal movement, and forks of the stacker, as well as the detection of accurate arrival at the bin and whether there are goods in the bin (Alizadehsalehi & Yitmen, 2023). The structure of the automated warehouse control system is shown in Fig. 5.

Figure 5 Structural block diagram of automated warehouse control system.

Design of improved Tabu Search algorithm

In this article, an improved Tabu Search algorithm is designed to solve the optimization problem of space allocation. The pseudocode for the improved algorithm is as follows:

Input: Functions at the port of entry and exit.

Output: Objective function of warehousing efficiency.

Batches now = seed.seed(C, X, Y,A).

Batches now = Batches now[1:].

if ratio * C <= (batches now[−1]) <= C:

batches.extend(batches now).

else:

Orders left.extend(batches now[]).

del batches now[]

batches.extend(batches now).

if ratio * C <= orders item number() <=

Batches.append ().

if orders item number(,) < ratio *

Orders left.extend().

orders = orders left.

if orders item number(orders) <=

batches.append(orders).

The candidate solution set may be equal to the neighborhood solution set or belong to the neighborhood solution set, that is, as long as the candidate solution set itself is a proper subset of the neighborhood solution set, the algorithm may not correctly select the optimal solution of the neighborhood solution set. But this situation will still reduce the search time. A better solution may still be obtained after many iterations. When there is no better solution in the candidate solution set selected in this article, the best inferior solution in the current situation is selected as the initial solution of the next iteration.

While fully retaining the local depth search ability of the traditional Tabu Search algorithm and the good global depth search ability of the genetic algorithm, the two algorithms overcome their respective shortcomings when they are used together. The step flow of the improved Tabu Search algorithm is shown in Fig. 6.

Figure 6 Improved Tabu Search algorithm flow.

The number of goods that should be allocated is obtained through the dynamic programming algorithm. It can be seen that the core and difficulty of the algorithm lies in whether the objective function (penalty evaluation function) can meet the requirements of the algorithm.

The data center module is used to store and manage data. It is also a piece of system memory, but it is closed and does not allow other processes to access it. The configuration file was called once during initialization. A configuration file is a file containing all variables (Fardad & Ali, 2023). All variable data is read here. The tool can be used to write configuration files interfacially and intuitively, and the establishment of configuration files can effectively eliminate hard coding so that the modification operation does not need to be in the source file (Wei et al., 2019). Some important codes of the data center module are as follows:

Xml Document doc = new Xml Document();

doc.Load(System.App Domain.Current Domain.Base Directory +

config Full Name);

data Array = new Data[doc.Last Child.Child Nodes.Count];

for (int i = 0; i < data Array.Length; i++)

Data temp = new Data();

foreach(Xml Node clone Child in doc.Last Child.Child Nodes[i].Child Nodes).

switch (clone Child.Name).

temp.length_PLC =

ushort.Parse(clone Child.Inner Text);

catch

temp.length_PLC = 1;

data Array[i] = temp;

The read-write data module is used for realizing the functions of establishing a link, reading data, writing data, breaking the link, and the like (Ekren, 2020). First, call the method that can be linked to the PLC hardware device through the IP address and port number. Through this method, a socket object with a long connection will be created to link with the physical object. The subsequent read and write operations will use this connection object.

Experimental analysis

Experimental data

The data used in the experiment are all from a production line in the production workshop, which is a product assembly line. The weight of each part is measured, as well as the process arrangement and operation time measurement in the product assembly process. According to the daily production plan of the day, under the existing warehouse layout, the size of the automated warehouse, the parameters of the shelves, and the stacker are designed according to the number of parts required in a certain cycle (Rahman, Janardhanan & Nielsen, 2020). The existing warehouse layout is shown in Fig. 7.

Figure 7 Warehouse layout.

Data collection

(1) Collection of warehouse and facility equipment data

The large automatic warehouse is equipped with automatic shelves, tunnel stackers, chain conveyors and car Kunzi conveyors. Select the shelf used for storing assembly line parts in the warehouse to carry out the operation of goods in and out of the warehouse. The height of the shelf is 6.00 m. There are 20 goods compartments in four layers and five rows. The size of each unit goods compartment is 1×0.6×1.5m, and the maximum load is 1,000.00kg. After being palletized, they are stored on shelves, and one pallet can only be equipped with one compartment. The operator of the parts in and out of the warehouse is equipped with a small lane-type stacker. (2) Assembly production data collection

Workers are scheduled to work 8 h a day and plan to produce 1,250 units. Firstly, the structure, assembly process and process flow of this kind of assembly parts are analyzed. The production time of each assembly part is calculated, and the operation process of each production beat and production line balance is calculated (Li et al., 2023). ① Product structure analysis of assembly parts

Obtain the assembly BOM and WI manual from the production line Kanban, and arrange the relevant data as shown in Tables 1 and 2 (Li, Shen & Zhou, 2023).

Table 1 Assembly EOM table and related information.

Number	Code	Quantity	Individual weight	Number	Code	Quantity	Individual weight	
1		1	50 g	11	E01⌋10	1	20 g	
2	B0120	1	10 g	12	E01⌋20	1	50 g	
3	B0130	1	10 g	13	E01⌋30	1	20 g	
4	B0140	1	10 g	14	B01⌋40	4	5 g	
5	B01_50	1	5 g	15	E01⌋50	1	10 g	
6	B0160	1	5 g	16	E01⌋60	2	25 g	
7	B0170	2	5 g	17	B01⌋70	2	10 g	
8	B01_80	4	10 g	18	E01⌋80	1	50 g	
9	B0190	1	5 g	19	E01⌋90	1	20 g	
10	B01⌋00	1	5 g	20	B01200	4	10 g	

Table 2 Summary of parts and components required for daily production.

Number	Part code	Weight of parts	Quantity	Number	Part code	Weight of parts	Quantity	
1	B01⌋0	62.5 Kg	1,250	11	B01⌋10	25 Kg	1,250	
2	B0120	12.5 Kg	1,250	12	B01⌋20	62.5 Kg	1,250	
3	B01_30	12.5 Kg	1,250	13	B01⌋30	25 Kg	1,250	
4	B0140	12.5 Kg	1,250	14	B01⌋40	25 Kg	5,000	
5	B0150	6.25 Kg	1,250	15	B01⌋50	12.5 Kg	1,250	
6	B01_60	6.25 Kg	1,250	16	B01⌋60	62.5 Kg	2,500	
7	B01_70	12.5 Kg	2,500	17	B01⌋70	25 Kg	2,500	
8	B0180	50 Kg	5,000	18	B01⌋80	62.5 Kg	1,250	
9	B01_90	6.25 Kg	1,250	19	B01⌋90	25 Kg	1,250	
10	E01⌋00	6.25 Kg	1,250	20	B01200	50 kg	5,000	

② Analysis of assembly process flow

Based on the WI of the assembly, the assembly sequence and time of each process are given, and the relevant data are shown in Table 3.

Table 3 Each process and assembly time of assembly parts.

Basic actions	Measured time	Action description	Work that must be done in advance	
A	20 s	Bracket assembly	–	
B	25 s	Installation of front and rear bearings	A	
C	6 s	Tighten the bracket screws	B	
D	20 s	Assembly of transmission components	–	
E	8 s	Weld the circuit board	D	
F	10 s	Insert the front panel	C	
G	10 s	Insert the back panel	C	
H	12 s	Remote control assembly installation	E	
I	5 s	Labeling	E	
J	12 s	Final Assembly	F,G,H,I	
K	8 s	Test	J	
Total time T	136 s	–	–	

③ Determine the takt time

The takt time R of each product is calculated by producing 1,250 assemblies in a working time of 8 H/day.

Takt Time:

(1) R=TC=8×1,800s1,000

(2) Smin=SC

where: T is the working time/s, C is the production quantity, and Smin is the minimum number of stations.

The operation flow of each process is shown in Fig. 8.

Figure 8 Process flow chart.

Algorithm solution and analysis of cargo location assignment optimization model

When calculating the allocation scheme of assembly parts, it is assumed that the start time, acceleration and deceleration time, braking time, and picking time are all o during the operation of the stacker, that is, the stacker always runs at a uniform speed in the horizontal and vertical directions (Arbex Valle & Beasley, 2020). The parts warehousing platform is located at the same end of the shelf, and a coordinate system with the position of the goods near the warehousing platform as the coordinate origin is established. The X-axis direction is the length direction of the shelf. The Y-axis direction is the height direction of the shelf, and the Z-axis is the width direction of the rack.

The adaptability of the optimal candidate solution in this article is much higher than that of the historical optimal solution. If a certain number of candidate solutions are selected in the neighborhood of the current solution, the solution will be accepted by the algorithm regardless of whether it is in the Tabu list (Hosamo et al., 2023). If it is uncertain that a candidate solution of this type exists, replace the current solution by selecting a candidate solution that does not exist in the Tabu list, and add its corresponding move to the Tabu list (Ward et al., 2021). Through repeated iterations until the requirements of the stop criterion are met.

Before solving with the algorithm, determine the specific values of the algorithm parameters. The algorithm parameters are shown in Table 4.

Table 4 Algorithm parameter setting table.

Parameters of control	Meaning	
W1 = 0.4, W2 = 0.3, W3 = 0.3	Weight values of three optimization objectives for location-allocation	
Tmax = 100	Maximum number of iterations	
L = 7	Taboo table length	
Δx=1	Non-Tabu feasible move	
T = t + 1	Tabu search counter	
n = 20	Number of parts	
alpa = 0.1	Normalization parameter	
NIND = 100	Population size	
GGAP = 0.9	Generation gap of genetic algorithm	
gen = 0	Various swarm initial evolution iteration counter	
Gen = 0	Optimal individual preserving algebraic counter	
Maxgen = 10	The optimal individual keeps the least number of generations	
Pc = 0.7	Crossover probability of chromosome individuals in population	
Pm = 0.7	Mutation probability of chromosome individuals in population	

Comparison of algorithm solution results

Input the pseudo-code in the MATLAB software and run the code to solve the location-allocation scheme, and finally get the sorting sequence of parts. According to the pre-set matrix definition, get the location coordinates of each part on the shelf through the sequence. At the same time, the iteration graph of the traditional Tabu Search algorithm is obtained by running the MATLAB software. The iterative results of the algorithm are shown in Fig. 9.

Figure 9 Tabu search algorithm iteration results.

After using the improved Tabu Search algorithm, that is, Tabu Search improves the genetic search population so that the performance of the algorithm has been significantly improved. The value of the objective function is 0.107 < 0.1381 (the result of the traditional Tabu Search algorithm). The algorithm has good convergence. The strategy of Tabu Search to improve the population makes the algorithm search for a considerable proportion of individuals in the population, which can effectively promote the population to develop in a better direction and make the results of the algorithm tend to the optimal solution.

Comparative analysis of storage location-allocation schemes and algorithms

The detailed results of the parts location-allocation optimization scheme of the assembly parts are shown in Tables 5 and 6.

Table 5 Location coordinates of goods location-allocation by traditional Tabu Search algorithm.

Number	Codes	Coordinates	Number	Codes	Coordinates	
1	B01_10	(3, 1, 1)	11	B01_110	(3, 3, 1)	
2	B01_20	(3, 4, 1)	12	B01_120	(1, 3, 1)	
3	B01_30	(2, 2, 1)	13	B01_130	(4, 4, 1)	
4	B01_40	(2, 5, 1)	14	B01_140	(4, 3, 1)	
5	B01_50	(3, 5, 1)	15	B01_150	(4, 5, 1)	
6	B01_60	(2, 3, 1)	16	B01_160	(4, 2, 1)	
7	B01_70	(1, 5, 1)	17	B01_170	(2, 1, 1)	
8	B01_80	(4, 1, 1)	18	B01_180	(1, 4, 1)	
9	B01_90	(1, 1, 1)	19	B01_190	(2, 4, 1)	
10	B01 100	(3, 2, 1)	20	B01 200	(1, 2, 1)	

Table 6 Location coordinates of improved Tabu Search algorithm for location-allocation.

Number	Number of subrogation location-allocation	Delegate location assignment	
Code	Coordinates	Code	Coordinates	
1	B01_10	(1, 3, 1)	B01_110	(3, 2, 1)	
2	B01_20	(4, 1, 1)	B01_120	(2, 2, 1)	
3	B01_30	(2, 5, 1)	B01_130	(1, 5, 1)	
4	B01_40	(4, 2, 1)	B01_140	(1, 2, 1)	
5	B01_50	(4, 5, 1)	B01_150	(4, 3, 1)	
6	B01_60	(2, 3, 1)	B01_160	(3, 1, 1)	
7	B01_70	(4, 4, 1)	B01_170	(3, 5, 1)	
8	B01_80	(2, 4, 1)	B01_180	(3, 3, 1)	
9	B01_90	(3, 4, 1)	B01_190	(2, 1, 1)	
10	B01 100	(1, 4, 1)	B01 200	(1, 1, 1)	

The coordinates of the cargo allocation scheme obtained by the traditional Tabu Search algorithm and the improved Tabu Search algorithm are substituted into the objective function to obtain the objective function value. (1) Warehouse-in and warehouse-out efficiency of assembled parts

The coordinates of the cargo allocation scheme obtained by the traditional Tabu Search algorithm and the improved Tabu Search algorithm are substituted into the objective function to obtain the objective function value. (2) Storage efficiency of assembled parts

(3) T1(x,y)=2×∑mx=1⁡∑ny=1⁡(Dxy×txy)

(4) Dxy=Dxy′×Volum

(5) txy=max{(x−0.5)×Lvx,(y−0.5)×Hvy}

Substitute the weight Dxy of the parts, the running speed vx=1.2m/s,vy=1m/s of the stacker and the coordinates of the parts in Table 2 into the formula to calculate the efficiency T=1629<8h of parts in and out of the warehouse. (3) Shelf stability

Equivalent center of gravity Gx in the overall horizontal direction of the shelf is as follows:

(6) Gx=∑mx=1⁡∑ny=1⁡Dxy×(x−0.5)×L/∑mx=1⁡∑ny=1⁡Dxy

Equivalent center of gravity Gy in the vertical direction of the overall shelf is as follows:

(7) Gy=∑mx=1⁡∑ny=1⁡Dxy×(y−0.5)×H/∑mx=1⁡∑ny=1⁡Dxy

The equivalent center of gravity GTS=(3.31,2.75), G=(2.18,0.92)are obtained by substituting the shelf size L=4m,H=6m, the weight of the parts and the location-allocation coordinates of the parts into Formula (6).

The objective function values obtained by the traditional Tabu Search algorithm and the improved Tabu Search algorithm are compared, and the comparison results are shown in Table 7.

Table 7 Solution results of traditional Tabu Search algorithm and the improved Tabu Search algorithm.

Objective function	Traditional tabu algorithm	Improved algorithm	Indications	
In and out storage efficiency	2,565.5 s	1,727 s	Reflect the storage operation capacity, the smaller the value, the better	
Overall equivalent center of gravity of the rack	(3.43, 2.86)	(2.39, 0.79)	Reflecting the shelf stability index, the closer the equivalent center of gravity is to (2.5, 0.5), the safer the shelf is	

According to the objective function value results of the two algorithms in Table 7, the results obtained by the improved Tabu Search algorithm are better than those obtained by the traditional Tabu Search algorithm. The operation time of the stacker is shorter. The overall equivalent center of gravity of the shelf is closer to the geometric center of gravity of the shelf in the horizontal direction and closer to the bottom position in the vertical direction.

In addition, when the traditional Tabu Search algorithm is used to solve the problem of cargo allocation, there is a possibility that the algorithm will converge in advance when the results are not mature. In the improved Tabu Search algorithm, the hybrid strategy of Tabu Search improved the diversity of the population and expanded the search area. To eliminate the accidental operation of the algorithm, the traditional Tabu Search algorithm and the improved Tabu Search algorithm are respectively run five times in the MATLAB software, and the multiple iterations of the two algorithms are obtained as shown in Fig. 10.

Figure 10 Tabu Search algorithm comparison test chart.

Stability coefficient refers to the ratio of the number of repeated runs of the algorithm and the number of times and the final run result is the same as the total number of runs of the algorithm. The higher the ratio is, the more stable the algorithm is. The stability factor is as follows:

(8) sta=nN

It can be seen from the trend of the fitness value curve and the final fitness value in the running iteration chart of the two algorithms in Fig. 10 that the fitness value of each of the five repeated operations of the traditional Tabu Search algorithm is different. The stability coefficient is 0, while the results of four of the five operations of the improved Tabu Search algorithm are the same. The stability coefficient is 0.8, which is much higher than that of the traditional Tabu Search algorithm.

Discussion

Through experimental analysis, it is verified that the Tabu list of the Tabu direct search can automatically block the current Tabu area filtered by the Tabu searcher in real-time, which can effectively avoid the model outflanking the direct search again. Then, build a benchmark that can release the area where the Tabu direct search is located to ensure the diversity of relevant information more effectively in the process of the second direct search.

According to basic knowledge, the Tabu machine learning system relies on the structural framework of neural network parallel search, combines the powerful potential of the Tabu machine learning system, avoids bypassing direct search and the diversity of genetic factors in genetic evolution, and promotes rapid global optimization. It fully retains the local organization depth direct search of the traditional Tabu machine learning system and the good global depth direct search potential of neural network database data. It also overcomes the shortcomings of traditional research content. The effectiveness of the algorithm is verified by an example, which can reduce the time and distance of picking to a certain extent and improve the efficiency of order picking. The research results can be used to guide the picking operation under the multi-block warehouse layout in practice and help to improve the timeliness of customer order fulfillment.

Conclusion

Based on the design of the automated warehouse, this article studies the automated warehouse, the supporting facilities and equipment of the automated warehouse, the distribution principle of goods location and the solution of the intelligent algorithm to the mathematical model. The main work includes: (1) The optimization goal is established to improve the efficiency of entering and leaving the warehouse and ensure the overall stability of the shelves. An improved Tabu Search algorithm is designed to solve the mathematical model of the optimization of storage allocation.

(2) A simulation experiment is designed to compare and analyze the validity and rationality between the model and the algorithm systematically.

Experiments show that the research content of this article solves the defect that the search is easy to fall into the loop, which gets a better solution and improves the calculation accuracy. In this article, based on the Tabu operator of the neural network, the specific process sequence of logistics warehousing is optimized. Experiments verify that the use of neural networks to optimize the storage operation sequence can effectively shorten the task completion time. To greatly improve the efficiency of goods in and out of the warehouse and operation, the principle of goods location-allocation is combined. The principle is transformed into an algorithm statement into an improved Tabu Search algorithm to reduce the running time of the stacker, optimize its picking efficiency, and achieve the purpose of fast in and out of the warehouse. It plays a decisive role in studying the operation procedures in the simulation system warehouse and how to allocate the warehouse storage.

With time, the enterprise’s production direction and the variety of goods will change, and the current applicable location allocation scheme may not adapt to future demand. Therefore, the location assignment algorithm proposed in this article has a high complexity. The complex model and objective function increase the amount of computation and occupy a certain amount of computer resources. The optimization of the algorithm’s efficiency will be the focus of future research.

Supplemental Information

Supplemental Information 1 OSHWH.

Supplemental Information 2 Program code.

Additional Information and Declarations

Competing Interests

Author Contributions

Data Availability

The authors declare that they have no competing interests.

Huajun Di conceived and designed the experiments, performed the experiments, analyzed the data, performed the computation work, prepared figures and/or tables, authored or reviewed drafts of the article, and approved the final draft.

The following information was supplied regarding data availability:

The program code is available in the Supplemental File.

The data is available at: https://cmp.felk.cvut.cz/t-less/.

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
