# Peer review of "Design and research of automated warehouse simulation platform based on virtual visualization framework"

_PeerJ Computer Science, doi:10.7717/peerj-cs.1809_

## Round 0.1 · original submission · Major Revisions

After receiving the reviewers' comments and my reading of the manuscript, I can recommend the manuscript for major revision. The main concerns are focused on the writing of the manuscript, format and layout, and some description about experiments. I suggest the authors to polish the writing.

Reviewer 1 ·

Basic reporting

This study is interesting and provides certain answers to the challenges permanently present in warehouses. However, it needs major improvements, which are commented on through suggestions and questions as follows:
1. The abstract needs to be substantially improved. The author has to provide an unambiguous, clear, and concise declaration of the problem and the motivation for the research, as well as its relevance to the targeted results.
2. It is necessary to improve the writing of the manuscript. I recommend paying more attention to the clarity of expression and readability, namely sentence structure, length, and others.
3. How much does the problem of location affect the efficiency of the warehouse? Why exactly was this problem chosen for algorithm development?
4. The introduction requires verification of the different related numerical indicators pointed out in previous studies.
5. It is stated in the first paragraph of the introduction that the software contributes to increasing the efficiency of the warehouse. In this context, it should be stated in what scope.
6. The objectives of the research are not clearly defined.
7. It is not common practice in academic research to utilize photographs to explain how an activity proceeds to be carried out. Due to that, figure 1 needs to be generated.
8. Figure 3 is not adequately clarified. What is the relationship between the conveyor and the conveyor system? What are the two-way lines supposed to explain?
9. The paper requires significant improvement in all figures' quality.

Experimental design

The study implements an advanced Tabu Search algorithm, which can be beneficial if there is an explanation of how practitioners can apply the suggested approach to address real-world problems. Also, it is important to explain how the proposed method can be valuable for future research. Additionally, answers to the following essential questions could be valuable in this study:
1. In the section Experimental Data, the statement "the stations are reasonably allocated to make the production line reach a balanced state" is not clear. It is necessary to explain and connect it with the input data and the expected results.
2. It is essential to explain clearly how the advanced Tabu Search algorithm improves the traditional one.
3. Why was the Tabu Search algorithm selected? What are the limitations of the algorithm, and what is the reliability of the obtained results?

Validity of the findings

Practical findings discussed in numerous studies in warehouse management should be used to complement research. In this regard, it needs to provide answers to the following questions and suggestions in order to improve this research.
1. The authors need to discuss the limitations of the proposed algorithm as well as the limitations of the case study.
2. What are your recommendations for future investigations related to Tabu Search and warehouse management?
3. Future research directions are related to the improvement of simulation and automation. However, what specific research would be justified for the automation of warehouse processes?
4. How would practitioners benefit from your approach?

Additional comments

This research is not sufficiently supported by the practical gaps and challenges in the locational activities of warehouse management.

Reviewer 2 ·

Basic reporting

1. The abstract needs to improve. It should include the existing work and its shortcomings. The method that this manuscript proposed should be illustrated. The technical innovations should be strengthened.
2. Generally speaking, the introduction part should provide sufficient information on the research results, proposed solutions, and main contributions of recent years.
3. English level should be improved, and this paper still needs to correct the wrong words and expressions.
4. Demonstrate the algorithm using pseudo code preferred by the computer profession in the third section.
5. The number of references is small and should be increased to more than 25.
6. The text in Figure 2 to Figure 5 should be in a uniform format. Please correct them.
7. There is an error in formula 12, please correct it.

Experimental design

1. The abstract needs to improve. It should include the existing work and its shortcomings. The method that this manuscript proposed should be illustrated. The technical innovations should be strengthened.
2. Generally speaking, the introduction part should provide sufficient information on the research results, proposed solutions, and main contributions of recent years.
3. English level should be improved, and this paper still needs to correct the wrong words and expressions.
4. Demonstrate the algorithm using pseudo code preferred by the computer profession in the third section.
5. The number of references is small and should be increased to more than 25.
6. The text in Figure 2 to Figure 5 should be in a uniform format. Please correct them.
7. There is an error in formula 12, please correct it.

Validity of the findings

1. The abstract needs to improve. It should include the existing work and its shortcomings. The method that this manuscript proposed should be illustrated. The technical innovations should be strengthened.
2. Generally speaking, the introduction part should provide sufficient information on the research results, proposed solutions, and main contributions of recent years.
3. English level should be improved, and this paper still needs to correct the wrong words and expressions.
4. Demonstrate the algorithm using pseudo code preferred by the computer profession in the third section.
5. The number of references is small and should be increased to more than 25.
6. The text in Figure 2 to Figure 5 should be in a uniform format. Please correct them.
7. There is an error in formula 12, please correct it.

·

Basic reporting

Based on the design of the automated warehouse, this paper studies the automated warehouse, the supporting facilities and equipment of automated warehouse, the distribution principle of goods location and the solution of the intelligent algorithm to the mathematical model. The experimental results verify its effectiveness.

Experimental design

Although the manuscript has a certain advantages, but there are also some concerns should be addressed.
A. The abstract should be improved. Please pay attention to the structure of it. Generally speaking, the proposed method and the research achievements should be focused.
B. The introduction part is not satisfying. Briefly introduce the contents of each section in the introduction part.
C. The text in Figure 1 is not clear. Please correct it.
D. The number of references is not enough. Please add some recent references.
E.Is the formula 12 correct? Please check and revise it.
F. The language should be checked and improved. The author is suggested to polish the language to make it achieve the publishing standards.

Validity of the findings

The experiments can reflect the features of the proposed solution. But the resolution of figures need to be further improved.

Additional comments

NA

---

## Round 0.2 · Major Revisions

The original Academic Editor is unavailable so I have taken over handling this submission.

Thank the author for the efforts to improve the work. In this round, the reviewers proposed some suggestions on writing, experimental design, etc. Please continue to revise the article accordingly.

**Language Note:** The review process has identified that the English language must be improved. PeerJ can provide language editing services - please contact us at copyediting@peerj.com for pricing (be sure to provide your manuscript number and title). Alternatively, you should make your own arrangements to improve the language quality and provide details in your response letter. – PeerJ Staff

Reviewer 4 ·

Basic reporting

In general, this is a pretty nice paper. However, the paper still has many limitations that need to be improved before being considered for publication.

1. Using acronyms/abbreviations in title is very bad practice. Please avoid.
2. The literature review needs to be improved by the recent investigation/developments in the field of AS/RS technologies. For this purpose, the authors can add (at least) the following references to enrich the literature review section:
a) Efficient localization in warehouse logistics: a comparison of LMS approaches for 3D multilateration of passive UHF RFID tags. The International Journal of Advanced Manufacturing Technology, 120(7-8), 4977-4988.
b) Optimizing automated storage and retrieval algorithm in cold warehouse by combining dynamic routing and continuous cluster method. In AETA 2018-Recent Advances in Electrical Engineering and Related Sciences: Theory and Application (pp. 283-293). Springer International Publishing.
c) Development and optimization of automated storage and retrieval algorithm in warehouse by combining storage location identification and route planning method. In 2017 International Conference on System Science and Engineering (ICSSE) (pp. 600-605). IEEE.
d) Building management algorithms in automated warehouse using continuous cluster analysis method. In AETA 2017-Recent Advances in Electrical Engineering and Related Sciences: Theory and Application (pp. 1068-1077). Springer International Publishing.

Experimental design

To be able to apply in practice, the control system needs to deal with system uncertainty and external disturbance, the author should consider these factors in your system simulation to ensure the algorithm robustness

Validity of the findings

In conclusion, please highlight particular analysis and comparison results, what confirm existing knowledge. Future research should also be under consideration.

Additional comments

Authors need to clarify the above points before the paper can be considered for publication.

·

Basic reporting

Thanks for the submission. The paper aims to study the problem of warehouse goods allocation using mathematical modelling with 3D simulation. The novelty of the paper is the mathematical model with improved Tabu Search algorithm and 3D simulation software.

1. "Abstract" section:
a. It would be really helpful if the paper includes a bit more background on how the proposed solution fundamentally differs from existing approach. From the following paragraphs in the
b. From line 61: for "improved Tabu Search Algorithm" => improved how? is it a modified version catered towards the specific warehouse problem or it works really well for all inputs? Would appreciate more clarification on this. And the readers would appreciate to understand why do we need to use a different version from the original algorithm("reducing the size of the problem" is a bit vague here).

2. "Introduction" section:
a. the research contributions of intelligent warehouses are complete per last revision suggestion.
b. From line 136: would be appreciate more background on Tabu Search algorithm. Since it is the backbone of the warehouse modeling solution in this paper, basic information should be included: What problem does the original Tabu Search algorithm solve? Who proposed the "improved" version? Is there any case studies about the "improve" version done in the past? Even if it is not related to the objective of this paper. And what is the scope of the algorithm?(what type of problems can we use it to solve?)

3. "Related work" section:
a. same question as the last review: I failed to find the pseudo code in lines 202 - 229. There is a "Improved Tabu Search algorithm Flow" in Figure 6 but using a pseudo code version in the paragraph to explain how the algorithm is improved would be even better. An alternative is to use side-by-side comparison between the original and improved version.

4. Overall: please double check the grammar. I noticed there are several grammatical errors regarding the singular/plural use forms.

Experimental design

Same as previous review comments and the question raised in the "Basic Reporting" section, would be helpful to include more information on the improved Tabu Search algorithm: what is the motivation to improve it? what are the major changes?

and most importantly: how did you design the experiment to verify that the improved version is better? how confident is the 0.107 < 0.1381 value here? Is it possible to generalize the improved version for other problems or this is task specific.

Validity of the findings

The findings are reached by running mathematical models with real-world data. The premises of the experiment design and the limitation with the warehouse automation are identified by authors via domain knowledge and real-world examples.

For claims regarding the improved Tabu Search: needs to discuss the scope of the improved version. Does the proposed improved version work for any task. limitations of the algorithm would also appreciate a bit more clarification.

In the conclusion section, would be great to discuss how beneficial this proposed method to the general public or the warehouse automation field. There are some literature studies related to the problem mentioned in the introduction section but we do not know how practical this solution is.

Additional comments

Thanks for the opportunity to review the manuscript. Please address the aforementioned questions.

---

## Round 0.3 · Minor Revisions

Thanks for the authors' efforts to make the article improve. The authors have issued most problems. However, the reviewers still have some concerns regarding format, spelling, pseudocode, etc. Please refer to the comments and continue to revise the work.

Reviewer 4 ·

Basic reporting

The authors have successfully improved their work in this round. I therefore advise to accept this work after after correcting the following points:
- References need to be carefully checked for format, for example references [7-10] do not have the author's name in the current version.
- need to check spelling again before publishing.

Experimental design

The question related to this part was answered satisfactorily.

Validity of the findings

The question related to this part was answered satisfactorily.

·

Basic reporting

Thanks for submitting the revision. This version of the paper explores the allocation of efficient goods in the Automated Storage and Retrieval Systems and introduces an improved Tabu search algorithm that directly searches and stores results to prevent infinite searching using iterative updates.

1. The literature review added more information regarding past research and existing SOTA solutions on automated warehouse control systems with references.
2. Thanks for addressing previous comments regarding the Tabu Search Algorithm and the difference between the proposed solution and existing ones.
3. Sections 3.3 and 3.4 combined together could provide a clear picture of how the improved Tabu search algorithm is written in MATLAB code. Regarding my previous review comment: “...but using a pseudo-code version in the paragraph to explain how the algorithm is improved would be even better. ”, I think the audience would benefit more from a formatted pseudo code of the improved algorithm.
4. All typos are fixed. Thanks!

The revised paper maintains a clear structure and provides background and motivation for the automated warehouse control study.

Experimental design

The motivation to improve the Tabu Search algorithm and changes are clearly explained. The experiment shows the adaptability of the improved version.

Based on the analysis of the specific warehousing control problem, this experimental design is sound and can be used to prove the argued point.

Validity of the findings

The findings are reached by running mathematical models with real-world data. The premises of the experiment design and the limitation of warehouse automation are identified by authors via domain knowledge and real-world examples.

In conclusion, the findings are valid considering the problem and its scope.

Additional comments

Thanks for the opportunity to review the manuscript.

Please address the comment regarding the pseudocode for the improved algorithm.

---

## Round 0.4 · accepted · Accept

Congrats to the authors. Thanks for the efforts you made. This version successfully satisfied the reviewers. It now can be accepted.

Reviewer 4 ·

Basic reporting

I propose to publish this paper at this revision.

Experimental design

I propose to publish this paper at this revision.

Validity of the findings

I propose to publish this paper at this revision.

Additional comments

I propose to publish this paper at this revision.

·

Basic reporting

The comments regarding the pseudo-code of the Tabu Search Algorithm and the differences between the two versions are addressed. Thanks!

Experimental design

All questions have been addressed by the author in the review response.

Validity of the findings

The findings are considered valid as it is reached by experimental design using real-world data.

Additional comments

All questions have been addressed by the author in the review response.